# Role of Transmembrane Water Exchange in Glioma Invasion/Migration: In Vivo Preclinical Study by Relaxometry at Very Low Magnetic Field

**DOI:** 10.3390/cancers14174180

**Published:** 2022-08-29

**Authors:** Maria Rosaria Ruggiero, Hamza Ait Itto, Simona Baroni, Sandra Pierre, Jean Boutonnat, Lionel M. Broche, Silvio Aime, François Berger, Simonetta Geninatti Crich, Hana Lahrech

**Affiliations:** 1Molecular Imaging Center, Department of Molecular Biotechnologies and Health Sciences, University of Torino, 10124 Torino, Italy; 2BrainTech Lab, INSERM U1205, 38000 Grenoble, France; 3Grenoble Hospital, 38000 Grenoble, France; 4Aberdeen Biomedical Imaging Centre, University of Aberdeen, Foresterhill, Aberdeen AB25 2ZD, UK; 5IRCCS SDN SynLab, 80143 Naples, Italy

**Keywords:** glioma mouse model, glioma invasion, AQP4, AQP1, FFC-NMR, relaxometry, very low magnetic field, transmembrane water exchange

## Abstract

**Simple Summary:**

Since the pioneering work of Damadian (1971), the longitudinal T_1_-relaxation of nuclear magnetic resonance (NMR) has been reported to generate significant contrast between cancer and healthy tissues at low magnetic fields. However, low NMR sensitivity was a substantial obstacle. Fast field cycling NMR (FFC-NMR) can overcome this problem and is commercially available for physics/chemistry research since around the years 2000s. Herein, using FFC-NMR in vivo, we show that T_1_-relaxation measured at very low fields is sensitive to transmembrane water exchange, thus allowing the discrimination between glioma invasion/migration and proliferation. Aquaporins 4 and 1 are found to be upregulated in invasion/migration, indicating that water exchange modulates T_1_-relaxations in glioma, via these aquaporins action. Overall, results suggest that, by blocking the aquaporin functions, the T_1_-relaxation should decrease in invasion/migration glioma. Results also stipulate that the entire invasion/migration volume could be visualized by FFC imaging, noninvasively. This may impact the medical community since invasion/migration delineation remains challenging by any medical imaging modality.

**Abstract:**

This work shows that the longitudinal relaxation differences observed at very low magnetic fields between invasion/migration and proliferation processes on glioma mouse models in vivo are related to differences in the transmembrane water exchange basically linked to the aquaporin expression changes. Three glioma mouse models were used: Glio6 and Glio96 as invasion/migration models and U87 as cell proliferation model. In vivo proton longitudinal relaxation-rate constants (R_1_) at very low fields were measured by fast field cycling NMR (FFC-NMR). The tumor contribution to the observed proton relaxation rate, R_1_^tum^ (U87: 12.26 ± 0.64 s^−1^; Glio6: 3.76 ± 0.88 s^−1^; Glio96: 6.90 ± 0.64 s^−1^ at 0.01 MHz), and the intracellular water lifetime, τ_in_ (U87: 826 ± 19 ms; Glio6: 516 ± 8 ms; Glio96: 596 ± 15 ms), were found to be good diagnostic hallmarks to distinguish invasion/migration from proliferation (*p* < 0.01 and 0.001). Overexpression of AQP4 and AQP1 were assessed in invasion/migration models, highlighting the pathophysiological role of these two aquaporins in water exchange that, in turn, determine the lower values in the observed R_1_ relaxation rate constant in glioma invasion/migration. Overall, our findings demonstrate that τ_in_ and R_1_ (measured at very low fields) are relevant biomarkers, discriminating invasion/migration from proliferation in vivo. These results highlight the use of FFC-NMR and FFC-imaging to assess the efficiency of drugs that could modulate aquaporin functions.

## 1. Introduction

Glioblastoma multiforme (GBM) is a highly lethal tumor entity characterized by tissue heterogeneity and, more specifically, by an invasive phenotype [1]. Despite considerable research efforts that have led to the development of innovative surgical techniques or radiotherapy and chemotherapeutic drug designs, the survival rate of patients diagnosed with GBM remains very poor [2]. One reason for glioma recurrence and treatment failure is due to the glioma cells remaining after surgical resection, which have the ability to both migrate and invade peritumoral regions while being invisible to the neurosurgeon [3]. Making these cells visible is therefore a major clinical need for neurosurgeons to optimize glioma resection. Many efforts are currently devoted to accurately delineating the peritumoral regions, also exploiting peculiar characteristics of the microenvironment [4,5]. Additionally, peritumoral regions are known to promote cell invasion, a mechanism considered to be one of the leading causes of death from cancer. One admitted hypothesis to cure glioma is to preferentially target these regions, where specific anti-invasion therapies could be more sensitive and efficient.

In particular, a priority in the fight against GBM is to develop new imaging modalities that can diagnose and quantify the effects of treatment drugs, since biopsy, which remains the gold standard for glioma phenotype characterization and grading, is rarely used for brain peritumoral regions. Such imaging technologies are challenging to design and the pathophysiological characteristics of the peritumoral area are not well known and remain poorly described, leading to recurrent modifications of the GBM classification with changing criteria [6].

Glioma invasion processes are highly orchestrated, involving multiple cross-interactions between genotypic and metabolic parameters. These complex interactions pose exciting challenges, with current results indicating new research directions for treatment. Facing the complexities of the microenvironment, interdisciplinary approaches based on concepts or tools from physics, chemistry, or engineering sciences provide several possibilities for the design of complementary and efficient diagnostic tools, therapeutic monitoring, and new targeted treatments.

Amongst the imaging modalities used in clinics, magnetic resonance imaging (MRI) has a privileged position in neuroscience since it is the only noninvasive diagnostic tool to cover the entire brain volume, and its underlying physics are a rich source of information that can be accessed thanks to the flexibility offered by radiofrequency (RF) pulse sequences. Many multiparametric MRI methods are now available, linking intrinsic or extrinsic MRI parameters to specific metabolic or functional features of the tumor tissue. In the case of glioma migration and invasion, two MRI methods have been proposed and are currently integrated in clinical investigations: (i) fluid-attenuated inversion (FLAIR) sequence, which nulls signals from the cerebrospinal fluid (CSF) and generates contrast based on both T_2_ and T_1_ relaxation time, and (ii) diffusion tensor imaging (DTI), a contrast mechanism which is based on the translational motion of water molecules. The FLAIR sequence is currently used post-neurosurgery to detect infiltrative tumor cells, which are commonly associated with the presence of white contrast on the margin of resection cavity and constitutes a bad prognostic indication. However, the specificity of FLAIR for the detection of tumor cell invasion remains a subject of debate. A recent work [7] shows that the FLAIR signal is rather linked to the pathological disturbances in the mechanisms of CSF exchange due to an encapsulation of the resection cavity by tumor cells. DTI, however, can quantify changes in the fractional anisotropy parameter and provides fiber-tracking imaging capabilities useful to detect modifications in the white matter (WM) microarchitecture due to the motion of tumor cells in the WM tracts [8].

FLAIR and DTI have both improved the assessment of the spatial extension of glioma, and new related biomarkers are still being proposed that correlate to disease [9] or glioma classification [10], yet the mechanisms responsible for these contrasts sources are not fully understood. Neuronavigation systems for preoperative and intraoperative MRI were also proposed to improve glioma invasion detection for efficient surgery, but both fail to delineate tumor margins. Fluorescence-guided surgery with 5-ALA was also proposed to enable real-time tumor margin discrimination by means of protoporphyrin IX (ppIX) fluorescence emission in neoplastic cells. However, glioma tissues do not exhibit visible ppIX fluorescence if the blood–brain barrier (BBB) is intact. On this basis, any approach to detect the peritumoral region makes it mandatory to refer to techniques that do not need the use of contrast agents (CAs). Finally, label-free terahertz (THz) reflectometry imaging has been shown to be able to detect low-grade tumors with intact BBB, showing a great potential to visualize peritumoral regions, but the penetration depth of THz waves in tissues is limited to about 500 μm due to water absorption.

Facing these challenges, techniques based on nuclear magnetic resonance (NMR) without CAs should be considered as key to characterize peritumoral regions in clinics. Indeed, our recent work on the search for contrast mechanisms in cancer based on proton NMR relaxometry investigations at low (<200 mT) and very low (<2 mT) magnetic fields using fast field cycling (FFC) methods was considered a good starting point in this context, too. This NMR technology is the only one currently available that allows measurements of the longitudinal relaxation time (T_1_) over a range of low and very low magnetic fields with good NMR signal-to-noise ratio. FFC measures relaxation at a given magnetic field after polarizing the sample with a strong magnetic field to ensure a high NMR signal sensitivity. In practice, FFC-NMR experiments periodically switch the magnetic field quickly between a strong polarization field to a weaker relaxation field (also called evolution field), and finally to a signal detection field. This whole process is repeated while varying the exposure time and strength of the relaxation field in order to follow the evolution of the magnetic relaxation at a variety of fields and to obtain the nuclear magnetic relaxation dispersion (NMRD) profiles, i.e., T_1_ versus magnetic field intensity. This capability to measure on the same device the T_1_ relaxation over a wide range of magnetic fields (from few mT to several hundreds of mT), covering several decades of Larmor frequency, is unique and allows assessing the molecular dynamic of water by using mathematical models applied to the NMRD profiles. This is particularly useful for the slow dynamics that characterize biological tissues.

Our hypothesis is that water mobility changes in glioma invasion/migration processes. In fact, tumor tissue has unusual protein structure and concentrations that may influence the water motion and its molecular dynamics, as well as its extra- and intracellular water exchanges. Interestingly, FFC showed a good capability to discriminate cancer in term of aggressiveness [11] with the potential to characterize tumor margins [12] and response to treatment [13] and, more specifically, to discriminate invasion from proliferation [14]. The latter study validated a relationship between the relaxation-rate constant (R1 = 1/T1) at very low magnetic field and the pathophysiological processes of glioma on ex vivo tissues of experimental glioma mouse models. Indeed, by using relevant animal models of glioma with a robustly characterized invasion/migration phenotype, the NMRD profiles of ex vivo glioma tissues exhibited lower relaxation-rate constants at very low fields in invasion/migration models (Glio6 and Glio96) compared with the proliferation model (U87). Moreover, functional experiments on glioma cells allowed us to obtain more insight into the rapid transmembrane water exchange and the decrease in R_1_ at low fields, showing the relationship with hypoxia and with the H_2_O_2_ redox signaling pathway, which are major hallmarks of glioma invasion/migration. Moreover, FFC markers from ex vivo glioma were associated with aquaporin 4 (AQP4) expression, a water channel protein of cell membrane that has been reported as a hallmark of glioma invasion in many studies [15,16,17].

The capability of FFC to measure water exchange in vivo without contrast agents responds, therefore, to a real need in biomedical research and clinical studies. Indeed, the information related to transmembrane water exchange is involved in several physiological and molecular processes. This mechanism holds a great potential as a biomarker for several diseases, as recently reminded by Springer [18], and has recently received a growing interest in cancer diagnosis [11,19,20] and for the understanding of cancer mechanisms [14].

Several NMR and MRI methods have been proposed to report on transmembrane water exchange in vivo in clinical settings. These approaches are generally based on the properties of magnetic contrast agents [21] but, recently, two MRI methods that do not require the use of contrast agents were proposed to assess different physiological water exchange processes: FEXI (filter-exchange imaging) [22,23,24,25] and tDKI (diffusion-time-dependent diffusional kurtosis imaging) [26,27]. Both exploit translational water diffusion properties in restricted compartments by using pulsed gradient NMR (PGSE) technique [28], which is easy to implement in any clinical MRI equipped with strong gradients. FEXI and tDKI are noninvasive and thereby attractive, however their NMR signals are complex, making it difficult to interpret the calculated water exchange parameters. Robust in vivo validation is therefore needed, especially since the intracellular water lifetime in the human brain found by the FEXI method was found to be one order of magnitude larger than methods using Cas in rat brain [20,29,30].

Here, we want to determine whether the mechanisms responsible for water proton longitudinal relaxation at very low field in vivo and ex vivo are dominated by transmembrane water exchange mechanisms and to assess the specific role of AQP4 in case of invasion/migration. Our objective is to find out whether the FFC biomarkers that have been identified ex vivo maintain their ability to discriminate between glioma invasion/migration and proliferation also in vivo. Therefore, the herein reported work aimed to confirm that the differences observed ex vivo between the dispersion profiles of glioma cell lines still hold in vivo by acquiring NMRD profiles on the same glioma mouse models [14], and to measure the transmembrane water exchange by FFC methods as reported in ref. [11]. The NMRD profiles and transmembrane water exchange were measured in vivo using the wide-bore FFC relaxometer developed by STELAR (Mede, PV, Italy) [11] at the University of Torino, Italy. The three glioma mouse models U87, Glio6 [8], and Glio96 [31] were obtained by injecting cells in mouse leg muscle. This was because the FFC relaxometer was not equipped with a gradient system so that the spatial localization of the tumor was performed by using a dedicated radiofrequency (RF) transmitter/receiver solenoid coil that could only accommodate the mouse leg [11].

In addition to the Na^+^/K^+^ ATPase, which has been shown to modulate the transmembrane water exchange [11], the immunohistochemistry (IHC) of the aquaporins AQP4 and AQP1 was also investigated as these two water channel proteins have been described to play major roles in glioma [16,32]. Standing the clinical relevance of contrast agent-free mechanisms reporting on ongoing pathophysiological processes of glioma in vivo, the herein reported observations may be useful to further prompt the development of FFC-imaging scanners.

## 2. Material and Methods

### 2.1. Animals

All in vivo animal protocols were approved in accordance with the guidelines and regulations of the European Guidelines for the Protection of Vertebrate Animals (decree 87–848 of 19 October 1987), licenses (C3818510003) from the French Ministry of Agriculture and by the Italian Ministry of Health (authorization number 807/2017-PR).

Immune-deficient nude mice (AthymicNude-Foxn1nu, female, 5 weeks age, weighing 30 to 35 g) were purchased from Envigo (Harlan, Gannat, France) and were kept under specific pathogen-free conditions in the animal facilities at Clinatec (CEA Grenoble, France) under agreement number B 38 185 10 003, accredited by the Grenoble local ethics committee (Cometh committee) and at the Molecular Biotechnology Center, University of Turin (d.lgs 26/2014). The animal treatment protocol was approved by the Italian Ministry of Health (authorization number 807/2017-PR).

### 2.2. Cells

Glio6 and Glio96 are experimental human glioma cell lines developed at the BrainTech Lab, Inserm U1205 [31,33]. Short tandem repeat (STR) profiling was not performed; however, all the glioma cells were authenticated by mRNA next-generation sequencing. Cells were obtained from surgical glioma resections. The procedure was approved by the Biological Resource Center Ethics Review Board 38,043 Hospital of Grenoble which obtained the consent for study participation from the patient or their family.

### 2.3. Glioma Cell Culture

U87 glioma cells (ATCC HTB-14) were grown under normoxia in 20% O_2_ and 5% CO_2_ in DMEM supplemented with GlutaMAX (ref. 31331-018 (Gibco)) and 10% fetal bovine serum and penicillin–streptomycin (100 U/mL), while Glio6 and Glio96 glioma cells were grown under hypoxia in 3% O_2_ and 5% CO_2_ in untreated flask with DMEM/F-12 (1:1) + GlutaMAX, supplemented with BFGF and EBF growth factors (20 ng/mL). All cell incubations were performed at 37 °C and cell viability was assessed using the trypan blue exclusion method.

### 2.4. Glioma Mouse Model

Glioma mouse models were obtained by injecting glioma cells in Dulbecco’s PBS (100 μL) into the muscle hind-limb of female Athymic Nude Fox_n_1^nu^ mice aged 8 weeks (Envigo Laboratories) and weighing 30 to 35 g. The number of injected glioma cells were 7.5 × 10^5^ for the U87 model, and 4 × 10^6^ for the Glio6 and Glio96 models.

During MRI and FFC measurement, the mice were anesthetized with a mixture of 20 mg/kg tiletamine/zolazepam (Zoletil 100; Vibac, Milan, Italy) and 5 mg/kg xylazine (Rompun; Bayer, Milan, Italy). The mice temperature was maintained using a gel pad heated at 37 °C.

Tumor dimension was measured by MRI, and in vivo FFC was performed when at least the 50% of the mouse’s leg was invaded by the tumor.

### 2.5. MRI

MRI follow-up was performed to control the in vivo tumor volume growth and was carried out in a 1T horizontal magnet (Aspect Magnet Technologies Ltd., Netanya, Israel) at the University of Torino. T_2_-weighted images (T_2W_) were acquired using fast spin echo sequence (FSE) with a repetition time TR = 3000 ms and an echo time TE = 50 ms in an FOV 50 × 50 mm^2^ and a matrix of 168 × 160. A total of 11 slices of 1 mm thickness were acquired using an RF excitation of 90°.

The magnet was equipped with a gradient of strength 450 mT/m and a 35 mm diameter solenoidal RF coil used in emission and detection.

### 2.6. Fast Field Cycling NMR

FFC experiments were performed in vivo to acquire NMRD profiles as well as to measure transmembrane water exchange by applying the two site exchange (2SX) model to the apparent bi-exponential T_1_ relaxation magnetization measured at different magnetic fields [11]. The measurements were performed on the Stelar FFC-NMR relaxometer (Stelar S.n.c., Mede (PV), Italy), equipped with a 40 mm 0.5 T FC magnet and a dedicated 11 mm solenoid radiofrequency coil that hosts the entire mouse’s leg. The pre-polarized sequence (PP/S) was used. It operates in three sequential steps: first, a strong polarization field B_0_^P^ is applied for a duration t_p_, to increase the sample magnetization and the SNR when experimenting in low fields; then an evolution field B_0_^E^ is applied for a duration t_E_, to make the NMR signal sensitive to the proton longitudinal relaxation T_1_ of interest; and finally the detection field B_0_^D^ is applied to detect the NMR signal. FFC measurements consist of repeating this process with different values of B_0_^E^ and t_E_ to measure the T_1_ relaxation at each evolution field, to obtain the NMRD profiles, i.e., T_1_ or R_1_ = 1/T_1_ against either B_0_^E^ or the Larmor frequency v_0_^E^ (v_0_^E^ = γ/2 π B_0_^E^). Note that B_0_^D^ is as high as possible but must still set the proton Larmor frequency on resonance with the RF coil.

Here, proton Larmor frequencies of B_0_^P^ and B_0_^D^ were set at 14.5 MHz and 13 MHz, respectively, and for each B_0_^E^ the signal was obtained with 32 t_E_ values over a long time range (2.8 to 4 s, depending on the evolution field) to sample both the rapidly and slowly relaxing magnetization components.

Seven ν_0_^E^ strengths were selected logarithmically between 1 and 0.01 MHz (0.01, 0.02, 0.037, 0.07, 0.15, 0.39, 1). An additional measurement was also performed at 10 MHz using the same parameters, except that the field sequence was a nonpolarized sequence (NP/S) for which B_0_^P^ is not applied.

The relaxometer operated under complete computer control with an absolute uncertainty in the 1/T_1_ value of ±2%.

### 2.7. NMRD Profile Analysis

At each B_0_^E^ field, T_1_ = (1/R_1_) values were obtained by fitting the magnetization decay curves by a mono-exponential model. However, since FFC measurements are not spatially localized, the signal measured originated from a mixture of tumor and healthy tissues. Thus, we extracted the relaxation of the tumor (T_1_^tum^) using the following Equation (1):(1)$$R_{1\;=\;}\left(f_{\mathrm{tum}}\times {\;R}_{1}^{\mathrm{tum}}\right)\;+\;\left(1\;-\;f_{\mathrm{tum}}\right)\cdot R_{1}^{\mathrm{mus}}$$
where R_1_ is the relaxation-rate constant of the whole volume detected by the RF coil, containing both tumor and healthy tissues; R_1_^mus^ is the relaxation-rate constant of the healthy hind-limb, and f_tum_ is the tumor volume fraction calculated from T_2W_ MRI images by using the ITK-SNAP software.

A high resolution was needed to probe the evolution time and check for bi-exponential behavior; therefore, only 8 evolution fields could be investigated. This was not sufficient to use the standard 3-Lorentzian model, which requires 7 parameters. The NMRD profiles (R_1_^tum^ versus ν_0E_) were therefore analyzed by isolating the segment that appeared nondispersive at low field from the dispersive part, by checking the gradient of the profiles in a log–log plot. The points showing similar slopes were average to provide the dispersion, following a power-law model described by Equation (2):(2)$$R_{1}^{\mathrm{tum}}\left(\nu \right)\;=\;A_{P}\nu^{\;-\;\beta }$$
where the offset parameter A_P_ and the component β are both used as FFC biomarkers to discriminate between tissues. The transition frequency between the dispersive and nondispersive regime was set to half of the gradient response. This analysis was made using Matlab 2018a (The MathWorks, Natick, MA, USA).

### 2.8. Transmembrane Water Exchange

In a second analysis, the magnetization decay curves were fitted by the apparent bi-exponential two-site-exchange (2SX) model described by Equation (3), allowing to extract the long and short relaxation-rate constants R_1L_ and R_1S_, respectively, as well as their corresponding fractional parameter a_s_:(3)$$M_{z}\;=\;M_{0}\left\{1\;-\;\left[\left(1\;-\;a_{s}\right)*\exp\left(\;-\;t*R_{1L}\right)\;+\;a_{s}*\exp\left(\;-\;t*R_{1S}\right)\right]\right\}$$

This analysis was performed in order to measure the parameters of cell membrane water exchange rate constant.

Indeed, the three processing parameters (R_1L_, R_1S_, and a_s_) are linked to the intra (R_1IN_) and the extracellular (R_1EX_) relaxations and their volume fractions V_IN_ and V_EX_, respectively, and, more interestingly, to the physiological parameters of water exchange, namely, the kinetic exchange rate constants k_IN_ and k_EX_, which are the reciprocal of the cellular water residence times τ_IN_ and the extracellular τ_Ex_, respectively. More details are reported in reference [11]. Note that only four parameters have to be extracted since V_IN_ + V_EX_ = 1 and the preservation of mass imposes V_IN_ × τ_EX_ = V_EX_ × τ_IN_.

For this work, we set R_1EX_ to the relaxation-rate constant of Matrigel, which mimics the extracellular compartment [34]. From this, the three parameters R_1EX_, τ_IN_, and V_EX_ can be derived according to the 2SX model [18,35]. These values were obtained by a simultaneous fitting of the magnetization relaxation over an extended range of magnetic field strengths that provided a higher accuracy for parameter estimation.

All these data were analyzed using the Origin software (OriginPro 8.5.0 SR1, OriginLab, Northampton, MA, Levenberg–Marquardt algorithm, RRID:SCR_014212) with V_ex_ between (0.09–0.19) and (0.15–0.5), respectively, for healthy and tumor mouse hind limbs as described in [11]. These two values were set with the same values for all fitting procedure. R_1ex_ was set to the experimental value of the Matrigel, obtained at each B_0_^E^ field [11]. Note that the results here measure the intracellular lifetime τ_IN_ = 1/k_IN_, where k_IN_, also called k_io_ by different authors, is the unidirectional water efflux rate constant (the subscript « io » indicates from in to out).

### 2.9. Histology and Immunohistochemistry

Hematoxylin/eosin (HE) histology and IHC of Na^+^/K^+^ ATPase, Ki-67, AQP1, and AQP4 were performed on fixed glioma tissues embedded in paraffin of 3 μm thickness. The benchmark system (Roche diagnosis) and the aquaporin antibody diluted 1/1000 were used for the AQP1 and AQP4, the Anti-Ki-67—(clone 30-9, Roche, ref. 05278384001) for the Ki-67 and Abcam, ab76020 for Na^+^/K^+^ ATPase. For all samples, the IHC expression was revealed by the ultraview Kit-DAB (Roche diagnosis). Na^+^/K^+^ ATPase images were acquired under 20X using Zeiss Axio scan Z1 slides scanner microscope (Photonic Imaging Center—PIC GIN), while all the others were examined under a light microscope (Olympus BX51) using the magnification (×100) for HE, (×100 and ×200) for Ki-67 and the aquaporins, and (×100 and ×200) for Na^+^/K^+^ ATPase.

In HE images, nuclei of glioma cells were stained in blue while the immunopositivity in IHC of Ki-67, AQP1, AQP4, and Na^+^/K^+^ ATPase appeared brown. For each glioma model, three different glioma tissues (n = 3) were analyzed. For each mouse and each HE and IHC, 5 slides were stained, quantitative analyses were performed from a typical one, and the data derived from the model were averaged.

Staining intensity of HE, Na^+^/K^+^ ATPase, Ki-67, AQP1, and AQP4 was quantified by measuring the marked surface in % according to Equation (4):(4)$${\;A}_{{\mathrm{IHC}}^{+}}\%\;\;=\;\frac{A_{{\mathrm{IHC}}^{+}}}{A_{\mathrm{tot}}}\times 100\%$$
where A_IHC+_ is the marked cells area and A_tot_ is the total cell area in the analyzed histological image.

Areas were measured using ImageJ software (NIH, Bethesda, Maryland, USA) after intensity-based segmentation and thresholding. Quantitative analysis was performed on a typical slide of each glioma tissue. Values were given by mean value in %.

## 3. Results

### 3.1. R_1_ at Very Low Field and Power-Law Model Parameters: Biomarkers of Glioma Invasion/Migration

The NMRD profiles shown in Figure 1A correspond to tumor and hind limb muscle tissue, respectively. The tumor, obtained by the glioma cells transplantation in the mouse leg, is shown in the T_2W_-MR image (Figure 1B). Differences in NMRD profiles were observed between the invasion/migration (Glio6 and Glio96) and proliferation (U87) models at low fields (<1 MHz) but not at 10 MHz. These differences were quantified by the relaxation-rate constant at 0.01 MHz (17.10 ± 0.42 s^−1^, 14.18 ± 0.34 s^−1^ and 13.63 ± 1.33 s^−1^ for U87, Glio6, and Glio96, respectively) (Figure 1C), when the dispersion profiles were built up using T_1_ calculated by the mono-exponential fitting rather than analyzing apparently bi-exponential decays (see above).

In Figure 2A, R_1_ values of the tumor tissue (R_1_^tum^) were derived by applying Equation (1).

In this case, significant differences in the relaxation rate (R_1_^tum^) were observed, especially at proton frequency ν_0_^E^ < 0.1 MHz. As shown in Figure 2B, R_1_^tum^ at 0.01 MHz were 12.26 ± 0.64 s^−1^, 3.76 ± 0.88 s^−1^, and 6.9 ± 0.64 s^−1^ for U87, Glio6, and Glio96, respectively, with *p* < 0.001 and <0.01.

By applying the power-law model of Equation (2), in addition to the parameter A_p_, which corresponds to relaxation-rate constant at very low field, we obtain the β parameter which reflects R_1_ dispersion related to water molecular dynamics. These two parameters have both lower values in Glio6 and Glio96, confirming higher dynamics of molecular water compared to U87 (Figure 2C,D), a behavior that is in agreement with the previously reported ex vivo observations, supporting the view that they may act as potential biomarkers of invasion/migration [14].

### 3.2. AQP1 and AQP4 (But Not Na^+^/K^+^ ATPase): Hallmarks of Invasion/Migration

Differences between U87 and Glio models were also observed by IHC (Figure 3A,B). Indeed, a higher Ki-67 expression (a marker of cell proliferation) was observed in U87 compared with Glio6 and Glio96, although not statistically significant, and an AQP4 overexpression (a marker of glioma invasion) was observed in Glio6 and Glio96 compared with U87 (*p* = 0.06 and *p* < 0.01, respectively), a finding that corresponds to the phenotypes of these three glioma models obtained with cells implanted in the leg muscle (U87 as proliferation and Glio6 and Glio96 as invasion models). In addition to our previous findings, the AQP1 was also found to be upregulated in Glio6 and Glio96 models by comparison to U87 (*p* < 0.5 and *p* < 0.01, respectively).

However, the IHC of Na^+^/K^+^ ATPase exhibits significant heterogeneities within the tumor as well as within the tumor group (Figure 4). Mean values were found around 22% for U87, 19% for Glio6, and 13% for Glio96 but with high standard deviations of 13, 27, and 15, respectively. These values are wide-ranging, therefore suggesting that Na^+^/K^+^ ATPase cannot discriminate between the three glioma groups.

### 3.3. Transmembrane Water Exchange: Determinant Role in Invasion/Migration Processes

Interestingly, despite that ex vivo and in vivo relaxation-rate constants are quite different (12 s^−1^ at 0.01 MHz for Glio6 ex vivo [14] against 4 s^−1^ in vivo here), the results show that relaxometry at very low field is sensitive to tumor features and malignancy in vivo as well as ex vivo. By applying the 2SX model of Equation (3) (illustrated in Figure 5A) to the apparent bi-exponential magnetization recovery curves (see typical example in Figure 5B), the intracellular water lifetimes τ_in_ were obtained and their values are reported in Figure 5C. It was found that the shortest values are for Glio6 and Glio96 (0.516 ± 0.008 s and 0.596 ± 0.015 s), intermediary for U87 (0.826 ± 0.019 s), and longest for healthy muscle tissue (1.190 ± 0.025 s). This later value is used here as a reference of healthy tissue in vivo. As expected, invasion/migration are characterized by lower values, and to our knowledge, this is the first time that τ_in_ are measured in glioma tissues in vivo comparing invasion/migration to proliferation models. The observed decrease in intracellular water lifetime is consistent with the in vitro observation of glioma cell pellets, although, as expected, quantitative values were found to be five to ten times greater in vivo than in vitro [14]. Overall, the behavior observed for the kinetics of water inter-compartmental exchange is consistent with the upregulation of the AQP1 and AQP4 (Figure 3). Indeed, the obtained results showed that the τ_in_ decrease and the AQP1 and the AQP4 overexpression are hallmarks in the case of invasion/migration. While the role of the AQP1 in glioma invasion/migration was studied by few groups [32,36,37], the role of the AQP4 is well established, as recently reviewed by Vandebroek et al. [16]. In the present study, new evidence is robustly demonstrated by using relevant mouse models of glioma invasion/migration.

The role of the enhanced transmembrane water exchange in invasion/migration is now well demonstrated, and the results strongly suggest that the rapid transmembrane water exchange could be considered as a reliable indicator of aggressive cancer characterized by a high cell invasion/migration.

The extracellular volume fractions were also obtained from the 2SX model analysis and were found to be larger in Glio6/Glio96 compared with U87 (Figure 5D). This difference was significant (*p* < 0.05) and should be consistent with the phenotypes of the three glioma models. Indeed, the extracellular compartment volume tends to be larger, probably to facilitate glioma cells to migrate, and smaller in the case of high proliferation, where cells tend to occupy the whole space as much as possible. Moreover, as previously reported [11], the data from this study confirm the significant contribution of the transmembrane water exchange mechanism to the longitudinal relaxation at very low magnetic fields. This in vivo result highlights again that water exchange rates across the cellular membranes in cancer can be assessed by a direct measurement of longitudinal relaxation-rate constants at very low field. The relationship between water exchange and relaxation as determined at low magnetic fields is of high interest and paves the way to the design of new therapeutic strategies. From the present study, one would suggest that, by blocking the AQP4 function selectively, a decrease would be expected in the rate of invasion/migration processes. Indeed, the AQP4 as a modulating agent has been already suggested in glioma therapy [16].

## 4. Discussion

Our work focused on the use of relaxometry at very low field to characterize glioma in vivo. The rationale of this work was based on the relationship between water molecular dynamics that characterize biological tissues and parameters of NMRD profiles. In particular, it is based on relaxation mechanisms at low and very low magnetic fields that have been described as sensitive to the physiological parameters of the transmembrane water exchanges. In addition, thanks to FFC-NMR, measurements of transmembrane water exchange parameters were realized. The method used does not require contrast agent administration, which is relevant for in vivo applications in preclinic and, more interestingly, in clinic. Our task was to assess the role of transmembrane water exchange as a biomarker of invasion/migration processes in this kind of tumor, in vivo. Glioma was chosen because it is reported as a paradigmatic model of cancer invasion/migration. Thanks to the large- bore FFC relaxometer prototype, in vivo FFC measurements were possible on tumor xenografts at the leg mouse. Unfortunately, the system was not suitable for measurements on the mouse brain. However, no significant difference to a preliminary study where FFC data were acquired ex vivo for U87 tumors implanted in intracerebral was observed. This finding allows us to assume that the phenotypes of both models (intracerebral and muscle leg) are very similar. This result was also confirmed by HE histology and Ki-67 and AQP4 IHC which both showed the analogous features in the two models despite differences in histology quantification between ex vivo and in vivo [14]. The absence of magnetic gradients in the FFC system made it impossible to spatially localize the volume of the tumor, but this limitation was circumvented using tumor masses occupying >50% of the total tissue, in order to match tumor volume to the sensitive volume of the RF coil. Interestingly, our values were found to be within the same order of magnitude as those measured in vivo on glioma patients by FFC imaging at Aberdeen. Indeed, at 10 KHz, R_1_ values were found equal to 3.8 s^−1^ (Glio6), 6.9 s^−1^ (Glio96), and 12.3 s^−1^ (U87) versus 5.5 s^−1^, 7 s^−1^, and 8 s^−1^ measured by FFC imaging at Aberdeen in three glioma patients (data collection in progress and not yet published). This first proof of concept in vivo encourages the development of preclinical FFC relaxometers to study orthotopic mouse glioma models. In our system, one solution is to use an RF surface coil with diameter of about 5 mm that can be placed on the ipsilateral hemisphere (where the tumor is implanted) and moved to the contralateral one, in order to compare tumor and healthy FFC signals. The realization of such an RF system is not trivial due to the very small diameter of the magnet bore of 40 mm, representing a significant limitation to place the mouse head on the magnet center. Overall, developments of FFC imaging adapted to preclinical studies are required and necessitate magnets with large bore from 100 mm to 20 mm.

While the A_p_ and β parameters of the power-law model were obtained and shown to be consistent with the investigated tumor phenotypes, the correlation times, which provide information about water molecular dynamic regimes, could not be assessed in this study because of the low number of tested evolution fields. This was due to a limitation in the experimental time, because the 2SX water exchange model, to perform the bi-exponential analysis of the magnetization decay, requires a high number of evolution fields. Thus, to limit the in vivo acquisition duration to less than 20 min, the number of experimental B_0E_ values was limited to eight, and thereby it was not suitable to apply the model-free approach [38] to the NMRD profiles accurately. Of course, we specifically privileged in vivo transmembrane water exchange measurements, which is a physiological parameter that attracts a growing interest in biomedicine [39], and several approaches for its quantification have been proposed [11,21,22,27,40]. The intracellular water lifetime (τ_in_) was found to be relevant in glioma characterization, as it was found to be shorter in invasion/migration glioma tissues than in the model describing the proliferation process. In general, the in vivo τ_in_ values of invasive glioma were found to be shorter than those measured by the same FFC technique in breast cancer [11], a result that could be associated with the high aggressiveness of Glio6/Glio96 tumors.

In this work, τ_in_ determination from FFC measurements was not possible in healthy brain, thus the comparison was made with the values calculated for muscle tissue. If our glioma values are compared to those of healthy rat brain in vivo, measured by others with methods that exploit paramagnetic contrast agent (CA) properties, one finds τ_in_ values of approximately 0.55 s [39]. Such low values arouse a concern about our data and hypothesis. Rigorously, one may consider that a comparison to rat brain would be not well set since Glio6 and Glio96, as well as U87 cell lines, were derived from surgical fragments from human glioma. Their transmembrane water exchange should therefore be compared to those from healthy human brain tissues. In this context, one may refer to the greater values obtained from a cultured cell system of neurons and astrocytes that were reported to be of equal to 0.75 s and 0.57 s, respectively [41]. Recently, τ_in_ calculation of normal human brains in vivo were derived from the apparent exchange rate (AXR) measured by FEXI method [23]. Surprisingly, the calculated τ_in_ values (τ_in_ = 1/k_io_ = 1/(fe × AXR), where fe is the extracellular water fraction ≈ 0.19 in healthy brain [30], were found equal to (1/0.19 ≈ 5 s and 1/0.14 ≈ 7 s in white matter (WM) and grey matter (GM), respectively, which is about ten times greater than those determined in rat brains in vivo [30]. In fact, the authors mentioned that more work is still needed to further understand the physiological basis of the apparent fast and slow diffusion components in brain tissue and the exchange between these two water diffusion pools. A clear limitation of this study is the lack of direct physiological validation of the water exchange measured here. Indeed, these values are higher than expected from our investigations but remain lower than those estimated from the ASL (arterial spin labeled) method where the water transport time through the neuronal membranes was evaluated to be of the order of several tens of seconds [40]. In addition, a discrepancy was found when comparing τ_in_ values obtained by FFC and by diffusional kurtosis imaging (DKI). Indeed, both methods were performed on the 4T1 breast cancer mouse model in vivo and, as reported, τ_in_ quantification by DKI (τ_in_ = τ_ex_/Ve, where in reference [26], τ_ex_ is the water exchange mixing time, not the extracellular water exchange and Ve the extracellular volume fraction) was found to be twofold lower (τ_in_ ≈ 350 ms when using Ve = 0.22) than that obtained by FFC of 680 ms [11]. This is also the case of the GL261 glioma model (τ_in_ ≈ 300–400 ms) compared to data of the present study (τ_in_ ≈ 500–800 ms). Moreover, this tendency to underestimate τ_in_ is observed when comparing the DKI data of healthy mouse (≈150–200 ms) to those already published on rat brains (≈500 ms) [30]. All these differences provided by different methods with or without CA are significant and show how difficult it is to measure τ_in_ precisely and accurately in vivo. These differences claim the need for cross-validations that should compare τ_in_ measurements in vitro on the same cell culture and in vivo on same animal models as well as on human volunteers. However, the most important point is the different τ_in_ values found in this study on different models of migrating and proliferating glioma using the same detection technique (FFC-NMR). Evaluation and comparison of τ_in_ measurements under drug effects or under physical stimuli should provide further validation.

Overall, despite the fact that measurements on normal brain tissue were not possible, we are confident in our results since they are consistent with the work performed on 10 glioma patients [19]. Indeed, the mean AXR value was found to be equal to 2.9 s^−1^ in viable tumor, which is greater than in GM with AXR = 0.4 s^−1^ and in WM with AXR = 1.1 s^−1^, leading to τ_in_ estimation of ≈1.7 s, ≈10 s, and ≈4 s, respectively. Thus, assuming that τ_in_ values of normal brain tissues are on the order of several seconds, in vivo glioma values measured in this study are estimated to be 5 to 10 times lower, a difference that should be easy to detect.

The role of the AQP4 in migration/invasion processes is now well established and its expression was found to be overestimated in invasion/migration with higher intensity than the AQP1. In contrast with AQP4, there are only few works describing the role of AQP1 in glioma, and our work confirms its potential as a marker of glioma invasion/migration, with a function probably coupled to that of AQP4, as suggested by Isokpehi et al. [42]. Here, the specificity of the AQP4 in invasion/migration is further supported by analysis of the Na^+^/K^+^ ATPase IHC. This agrees with our previous work where we showed the sensitivity of the AQP4 to hypoxia and H_2_O_2_ redox signaling [14], which are connected to migration/invasion. Indeed, the expression of Na^+^/K^+^ ATPase, which is also known to play a major role in water exchange kinetics [18], was found to be heterogeneous and not discriminating the three glioma models (Figure 4), probably due to the major role of the AQP4 in accelerating water exchange mechanisms in glioma invasion/migration.

Finally, despite that R_1_ measurements are generally simple to obtain, the method herein described requires NMR/MRI instruments operating at very low fields or, more interestingly, FFC imaging. Such technologies are still not largely available because of significant limitations of FFC-NMR and FFC imaging for in vivo applications due to low signal-to-noise ratio, as well as to long acquisition times. However, this technology could benefit from the significant hardware and software developments that have emerged in MRI over the last decade, mainly focused on increasing NMR signal sensitivity, making stable NMR signals over time, and developing MRI scan acceleration methods.

## 5. Conclusions

Thanks to our robust mouse models of glioma invasion/migration (Glio6 and Glio96) and to the access to a wide-bore FFC-NMR, it was possible to carry out relaxometry measurements that allow assessing transmembrane water exchange on invasion/migration tissues in vivo.

To our knowledge, this is the first time that longitudinal relaxation-rate constants (R_1_) at very low field and the intracellular water lifetime (τ_in_) have been quantified in vivo in glioma invasion/migration and proliferation animal models. The results showed a clear decrease of τ_in_ values in the case of invasion/migration with respect to proliferation.

Overall, the in vivo results confirm the relationship between proton longitudinal relaxation at low field and water exchange mechanisms, in accordance with the early work of Koenig [43] and the recent achievements in breast cancer [11]. Indeed, the lower relaxation rates observed in vivo for both models of glioma invasion/migration, with respect to the proliferation one, are maintained in the ex vivo [14] studies involving the same tissue specimens. We underline that the sensitivity of the relaxation parameter to transmembrane water exchange mechanism at very low field is outstandingly relevant. This makes the relaxation at very low field a quantitative index of water exchange kinetics, a biomarker that should report on the ongoing metabolic activity in cancer [44]. Indeed, the herein-reported in vivo results are in line with the functional experiments performed on cell pellets in vitro, in which it was demonstrated the direct effect on R_1_ at low fields of two main pathophysiological processes of invasion/migration, i.e., hypoxia and H_2_O_2_ redox signaling. Furthermore, it was shown how the decrease of R_1_ was accompanied by the upregulation of AQP4 transporters.

Overexpression of the AQP4 in our glioma models of invasion/migration is confirmed by immunohistochemistry in this study, and the overexpression of the AQP1 was also detected in invasion/migration. IHC supports the view of a dominant role played by these two water channel proteins in glioma invasion, a conclusion which also has been suggested or demonstrated by several groups. These findings support the idea to use drugs that target AQP4 function directly, or both AQP4 and AQP1, or indirectly by targeting hypoxia and H_2_O_2_ redox signaling.

The in vivo experiments performed in this study were mandatory to definitively demonstrate our hypothesis, thus promoting FFC-NMR as a potential technology in deciphering glioma invasion/migration from proliferation. The results from this study may be relevant to accelerate the development of MRI scanners operating at very low field by FFC imaging and to make them available in clinical settings and research laboratories, as the visualization of invasion/migration remains a challenging task in medical imaging.

## Figures and Tables

**Figure 1 cancers-14-04180-f001:**
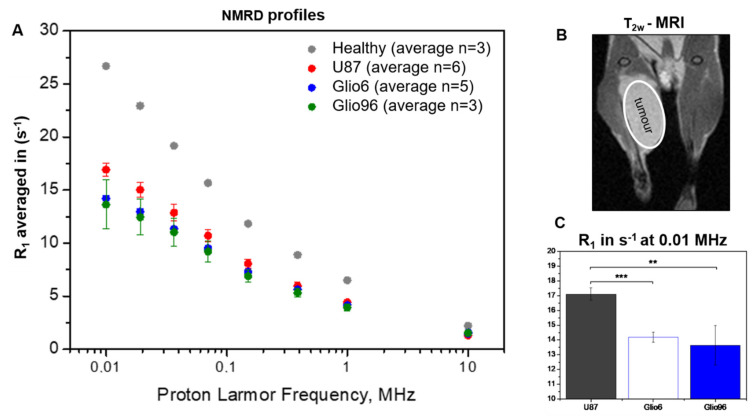
FFC-NMR characterization of glioma mouse models in vivo. (**A**): “In vivo” NMRD profiles of invasion (Glio6, Glio96), proliferation (U87) glioma models, and healthy hind limb muscle tissues. (**B**): Typical T_2W_-MRI showing the tumor region in the mouse leg when acquired by FFC-NMR in vivo (case of Glio6). (**C**): R_1_ values at 0.01 MHz observed for Glio models (invasion) and U87 (proliferation). Error bars represent the standard error of the experimental data (*** *p* < 0.001, ** *p* < 0.01).

**Figure 2 cancers-14-04180-f002:**
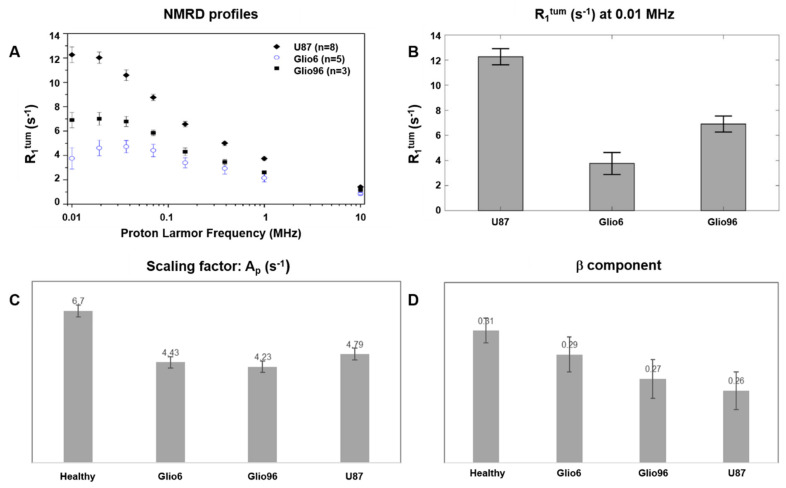
Analysis of relaxometric data of glioma tissue after signal muscle substraction. (**A**): NMRD profiles of glioma tissues showing a more marked discrimination between Glio models (invasion) and U87 (proliferation) at very low fields (Error bars report the standard error), and (**B**): their corresponding R_1_^tum^ quantifications at 0.01 MHz. (**C**): scaling factor (A_p_) and (**D**): component (β) parameters of power-law model, both confirming higher water molecular dynamics of Glio6 and Glio96 glioma tissues, compared to U87.

**Figure 3 cancers-14-04180-f003:**
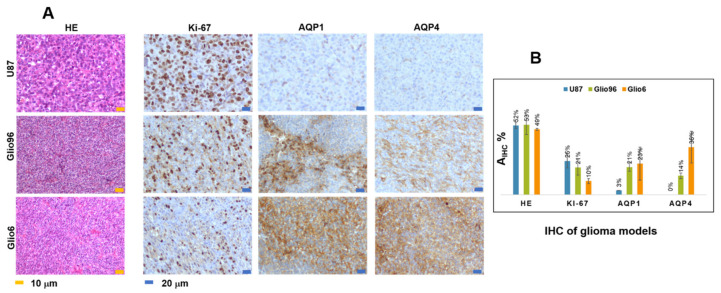
(**A**): Typical histological images: HE (×100) and IHC of Ki67, AQP1, and AQP4 of the three glioma models U87, Glio6, and Glio96 (×200). In HE histology, cell nuclei are stained in blue and in Ki-67, AQP1, and AQP4 IHC, the positive expressions are stained in brown. (**B**): IHC expression quantified by the parameter A_IHC+_%. Overall, IHC data confirm U87 as a proliferation glioma model (high Ki67 expression) and Glio6 and Glio96 as invasion model (low Ki-67 expression related to high AQP1 and AQP4 expression).

**Figure 4 cancers-14-04180-f004:**
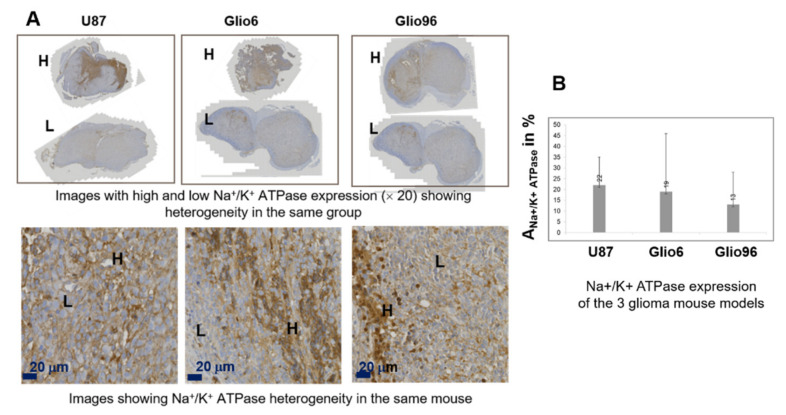
(**A**): IHC of Na^+^/K^+^ ATPase, stained in brown. All images were obtained from different mice in order to show the high heterogeneity of Na^+^/K^+^ ATPase expression within glioma groups and within each mouse. Top: For each glioma mouse model, typical images (×20) of two mice are presented: with high (H) and low (L) Na^+^/K^+^ ATPase expression. Bottom: images (×200) of the Na^+^/K^+^ ATPase, in which expressions appear heterogeneous within each image and for all the three glioma models. (**B**): IHC expression of Na^+^/K^+^ ATPase quantified by A_IHC+_% parameter. Standard deviation in the three models was found to be large, demonstrating a high Na^+^/K^+^ ATPase expression heterogeneity. No statistical differences were found when comparing Glio models to U87.

**Figure 5 cancers-14-04180-f005:**
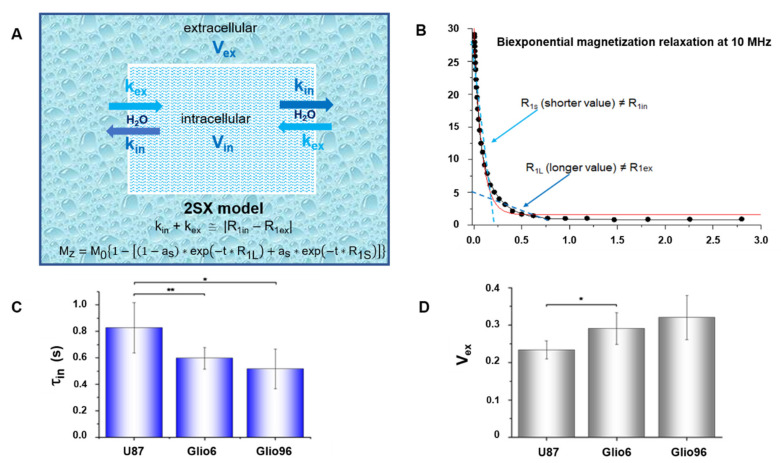
(**A**): Scheme and equations of the 2SX transmembrane water exchange model. (**B**): Typical bi-exponential of short (R_1S_) and long relaxation (R_1L_) of glioma tissue in vivo (case of Glio6 acquired at 10 MHz). Red and black continuous lines correspond to mono-exponential and bi-exponential fitting, respectively. (**C**): Mean values of the intracellular water lifetime (τ_in_) of the three glioma models, showing rapid water exchanges in case of invasion (Glio6 and Glio96). (**D**): Mean extracellular volumes (V_ex_). Larger V_ex_ are detected in the case of invasion (Glio6 and Glio96). Data are reported as mean ± SD (* *p* < 0.05, ** *p* < 0.01).

## Data Availability

The main data presented in this study are included in Figure 1, Figure 2, Figure 3, Figure 4 and Figure 5 and are available upon reasonable request from the corresponding author.

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
