# Peer review of "Role of Transmembrane Water Exchange in Glioma Invasion/Migration: In Vivo Preclinical Study by Relaxometry at Very Low Magnetic Field"

_cancers, 2022, doi:10.3390/cancers14174180_

Round 1

Reviewer 1 Report

Technically, the work seems be  fine however the Authors could better explain to the readers why low field technology has been used for this study. The Authors must remember that this journal is dedicated to a broad spectrum of readers; they may understand IHC studies; however, they do not understand technical explanations; there should be a few sentences more allowing to understand the purpose of low introduction of low field technology to this study. As was mentioned above all the reasoning seems to be carried out correctly, and the data from the experiment using relaxometry at a very low magnetic field are well supported by immunohistological analyzes. An important issue that can be discussed is the adequacy of the model selected by the Authors - why a xenograft glioma that achieves a paw size of 50% has been chosen? Such a large tumour could not be grown orthotopically, and I guess such a tumour was needed to obtain the correct measurement. It can be assumed that also that, such a large tumour will be inhomogeneous, it will have hypoxic and necrotic areas that can strongly disturb measurements by the Authors. I must admit that the Authors showed convincing data that they considered such problems in their experiment.

To sum up, a model was used in the work, which may seem a bit artificial, but provides valuable data for the design of the following experiments. I agree with the Authors that The results from this study may be relevant to accelerate the development of MRI scanners operating at the very low field by FFC-imaging and make them available in clinical settings and research laboratories, and may lead to the development of new methods of medical imaging. In my opinion, some improvements explaining technical aspects for the readers with biological background should be introduced in the Introduction and Discussion

Author Response

Dear Reviewer 1

First of all, thank you for your precious review and please,  find my responses in the document uploaded here.

Reviewer 2 Report

Authors have very elegantly addressed the limitations of current imaging technologies for brain tumours and introduced an emerging contrast-free technology Fast-Field-cycling NMR  for the identification of tumour localisation.

I only have some minor comments:

(1) Please state the exact strain of the animals used on page 4, line 183.

(2) Please state if the glioma cells used have been authenticated by STR profiling

(3) Please state if the Ki67, AQP1 and AQP4 are statistically significant between the different glioma models.

(4) Please include the IHC values for Na/K ATPase staining as a bar graph in Figure 4 with stats.

(5) Please highlight more clearly the magnification bar for Figure 4

(6) Do the authors have any thoughts on why the Na/K ATPase expression is so heterogeneous compared to the aquaporins?

(7) Could the authors also make a statement in their discussion on how to move this technology forward for preclinical studies in orthotopic brain tumour models as well as clinically?

Author Response

Dear reviewer 2

Thank you for your interesting revision and please find our responses in the document uploaded here
